# Impact of COVID-19 Pandemic on Flu and COVID-19 Vaccination Intentions among University Students

**DOI:** 10.3390/vaccines9020070

**Published:** 2021-01-20

**Authors:** Roberta Pastorino, Leonardo Villani, Marco Mariani, Walter Ricciardi, Guendalina Graffigna, Stefania Boccia

**Affiliations:** 1Department of Woman and Child Health and Public Health—Public Health Area, Fondazione Policlinico Universitario A. Gemelli IRCCS, 00168 Rome, Italy; roberta.pastorino@policlinicogemelli.it (R.P.); walter.Ricciardi@unicatt.it (W.R.); stefania.Boccia@unicatt.it (S.B.); 2Section of Hygiene, University Department of Life Sciences and Public Health—Università Cattolica del Sacro Cuore, 00168 Rome, Italy; marcomariani1990@hotmail.it; 3Dipartimento di Psicologia—Università Cattolica del Sacro Cuore, 20123 Milan, Italy; guendalina.graffigna@unicatt.it; 4EngageMinds HUB—Consumer, Food & Health Engagement Research Center, Università Cattolica del Sacro Cuore, 26100 Cremona, Italy

**Keywords:** COVID-19 vaccination, flu vaccination, students

## Abstract

Flu vaccination for the general population, and specifically for vulnerable subgroups, brings the potential to reduce the impact of the COVID-19 pandemic in terms of mobility, mortality, and hospitalizations. In Italy, flu vaccination is recommended to all ≥6 months of age, even if it is only free of charge for specific categories. We investigated the intentions towards flu and COVID-19 vaccinations from a sample of 436 Italian university students. Results of a web-based survey show that 77.52% of them were willing to get the flu vaccine and 94.73% were willing to be vaccinated against COVID-19 when available. We identified positive predictive factors to undertake flu vaccination as being a medical student, having undertaken a previous vaccination against flu, and having a high level of concern and perceived vulnerability to the COVID-19 pandemic. Reinforced public health activities might consider engaging university students a as possible “positive influencer” towards flu and COVID-19 vaccination programs.

## 1. Introduction

Flu vaccination represents one of the most effective strategies in order to reduce the healthcare, social, and economic impact of influenza [1]. Although influenza has a high impact in terms of mortality and morbidity in Italy, with about 7 million annual cases and an average annual mortality excess rate per 100,000 ranging from 11.6 to 41.2 [2], the vaccine coverage among older people and persons with chronic illnesses is low and very distant from the 75% recommended by the World Health Organization [3].

Since influenza and COVID-19 share similar symptomatology, especially in the first phase of the disease, a differential diagnosis between them will require tests, hospitalizations, and quarantine of symptomatic persons before obtaining the test results that could easily saturate the healthcare system capacity [4]. For this reason, there is an urgent need to implement flu vaccination for the target population [5]. In this context, some countries, Italy included, strongly recommend flu vaccination to all those people aged ≥6 months, even if it is only provided free of charge for specific risk groups of the population [6,7]. Nevertheless, over the last decades, opposition to vaccination has been amplified, and the phenomenon of vaccine hesitancy has arisen. This phenomenon is defined as a set of beliefs, attitudes, and behaviors (alone or in combination), exhibited by common people and, in some cases, also by healthcare professionals, in regard to their own or their children’s immunizations, causing a decrease of vaccine coverage and an increased risk of preventable disease epidemics [8]. In this context, the fear of potential damages after receiving the vaccine raises concern among people possibly leading to a reduction in the vaccine uptake [9]. In addition, self-perceived risk (both of influenza and COVID-19) may impact the willingness to get vaccinated [10]. Although the attitudes towards a future vaccine to prevent COVID-19 have been already investigated in the general and young population [11,12], evidence on the impact of COVID-19 pandemic on flu vaccination intentions in the young population is lacking. A special category of the young population is made up of university students, who are usually open-minded and capable to respond faster to public health issues [11].

With our study, we aimed to investigate the intentions of Italian university students regarding flu and COVID-19 vaccinations in the period of the COVID-19 pandemic, and the potential impact of concerns and perceptions about the COVID-19 pandemic on such intentions.

## 2. Methods

We conducted a web-based survey between 8 June and 12 July 2020, in the period immediately following the lifting of the Italian lockdown [13]. We administered an anonymous questionnaire on the intranet personal web page of undergraduate students attending one of the four campuses of Università Cattolica del Sacro Cuore. The University has four campuses located in the cities of Milan (Lombardy region), Brescia (Lombardy region), Piacenza-Cremona (Lombardy and Emilia-Romagna regions), and Rome (Lazio region). Therefore, there are four campuses in three different Italian regions. The faculties involved were Medicine and Surgery (Rome), Psychology (Milan and Brescia), Economy (Rome), Economy and Law (Piacenza-Cremona), Agricultural, Food, and Environmental Sciences (Piacenza-Cremona), Banking, Financial, and Insurance Sciences (Milan), and Education Sciences (Milan). The protocol of the study was approved by the Ethics Committee of the Policlinico Universitario A. Gemelli IRCCS and by the Internal Board of the University (since the voluntary and unpaid participation of students requires informed consent). The self-administered questionnaire included 30 items across three sections. Section 1 gathered students’ demographic information, such as age, gender, date of birth, faculty, year of study, and residence region. Section 2 included 7 items investigating their intention to get vaccinated against influenza and their opinions about a future COVID-19 vaccine. Section 3 investigated concerns and perceptions related to fear about an increase of COVID-19 cases, deaths, risk of getting infected, capacity to contain the spread of the virus, and their understanding of preventive measures. Eventually, we performed descriptive analyses for all variables, and survey-weighted logistic regressions to assess the influence of independent variables on each binary outcome investigated (with the results expressed as odds ratios (OR), 95% confidence interval (CI). In order to account for underrepresented groups in the student population, poststratification has been applied for adjusting the sampling weights. We considered *p*-values below 0.05 as statistically significant. We carried out all the statistical analyses using Stata software, version 14 (StataCorp LP, College Station, TX, USA).

## 3. Results

We collected a total of 436 questionnaires with a response rate of 78% (436/559). The median age of the students was 23.09 years (Interquartile range [IQR] 21.32–24.74), and females accounted for 70.41% of the sample. A total of 274 (62.84%), 88 (20.18%) and 74 (16.97%) students attend the Faculty of Medicine and Surgery (medical students, dentistry, and other healthcare profession students), Faculty of Psychology (Psychology students only), and Other (Economy, Economy and Law, Agricultural, Food, and Environmental Sciences, Banking, Financial and Insurance Sciences, and Education Sciences), respectively, with an overall higher prevalence of first-and second-year students (39.22%).

Almost eighty percent (77.52%) of the students (N = 338) were willing to undertake flu vaccination within the forthcoming vaccination campaign. With regard to a previous flu vaccination, 228 (52.29%) students, of whom 173 (75.88%) attend the Faculty of Medicine and Surgery, had received it at least once in their lifetime.

Concerning the opinions about a future COVID-19 vaccine, results show that 405 (92.89%) of students declared that they would feel more defended when the vaccine is available, and 399 (91.51%) believed that the COVID-19 vaccine will solve the emergency. Although a minority of students (N = 81, 18.58%) reported being afraid about possible adverse COVID-19 vaccine reactions, the majority (N = 413, 94.73%) were willing to be vaccinated against COVID-19 when available. Concerning perception and concerns about the COVID-19 pandemic, over 95% of the respondents (N = 420) reported understanding the relevance of lockdown measures adopted in the first wave of the pandemic in Italy, in terms of pandemic containment. Two hundred ninety-four students (67.40%) referred to fear about the increase in positive cases and 314 students (72.02%) were concerned about the reported number of deaths. In general, 169 students (38.80%) reported great concern about the containment of the pandemic in Italy, and around 60% of the students (N = 257) were willing to contribute more to face the pandemic. Concerning the students’ social life, 74.08% (323), 58.48% (N = 255), and 66.06% (N = 288) suffered from the impossibility of seeing friends, of seeing colleagues, and of attending the university, respectively.

In the multivariable analysis of predictors of the intention to undertake flu vaccination, medical students were more willing compared to psychology and other faculties together (adjusted (adj) OR 4.89, 95% CI (2.70–8.85)). A previous vaccination against flu was a significant predictor for repeating the vaccination (adj OR 5.82, 95% CI (3.27–10.37)).

The concern about the general COVID-19 pandemic (adj OR 1.13, 95% CI (0.99–1.30)), the fear about the increase in positive cases (adj OR 1.44, 95% CI (1.14–1.83)), and deaths (adj OR 1.35, 95% CI (1.05–1.74)) were predictors of the willingness to be vaccinated against flu. The understanding of the preventive measures was associated with the intention to get vaccinated against flu (adj OR 1.53, 95% CI (1.02–2.29)) as well as the willingness to contribute to efforts to control the pandemic increases, although the alpha did not reach <0.05 (adj OR 1.22, 95% CI (0.98–1.53)). Concerning the social life of students, the distance to colleagues, impossibility of attending university, and impossibility of seeing friends were associated with the intention to get the flu vaccine (adj OR 1.21, 95% CI (1.01–1.46); adj OR 1.20, 95% CI (1.01–1.44); and adj OR 1.31, 95% CI (1.07–1.60); respectively).

Furthermore, the willingness to undertake flu vaccination was associated with the intention to get vaccinated against COVID-19 (adj OR 9.58, 95% CI (4.78–19.19)), without differences among the faculties. Lastly, we report an association between being already vaccinated against flu, concerns about the increase in positive cases and concerns about the increase in deaths with the intention to be vaccinated against COVID-19 (adj OR 2.06, 95% CI (1.07–3.97), adj OR 1.58, 95% CI (1.18–2.10), and adj OR 1.68, 95% CI (1.24–2.66), respectively). The distributions of the selected covariates and adjusted ORs are shown in Table 1.

## 4. Discussion

In 2019, the WHO listed vaccine hesitancy as one of the 10 top health threats of that year [14]. In 2020, in the midst of the COVID-19 pandemic, vaccine hesitancy appears as an even more challenging health threat as it can compromise the effectiveness of any new potential vaccine [15,16] at the population level. In the past seasonal campaigns, in Italy, flu vaccine coverage lagged far from the values of at least 75% [17] as a minimum target and even farther from the 95% optimal target. In fact, after a peak in the 2005/06 season when 68.3% of the elderly population got vaccinated against influenza, a steady decline was reported, reaching 54.6% in the past 2019/20 season [18,19]. The flu vaccination coverage is even lower in the general population, reaching a 16.8% coverage rate in 2019/20 (3.1% in the age class 18–44 years) [19]. In our study, we investigated the intentions towards flu and COVID-19 vaccinations in a sample of university students, in Italy, that attend different faculties including Medicine and Surgery. The results show that the vast majority of the students are willing to undertake flu vaccination during the COVID-19 pandemic period. Therefore, despite the fact that, in general, students are a population that presents a low flu vaccination intention, the pandemic may have increased the willingness to vaccinate. It has already been observed, in fact, that an intention to vaccinate has reached almost 60% in the student population for the 2020–21 campaign [20], a figure higher than the average values generally observed at the national level. Our study shows a slightly higher willingness to vaccinate (77%) which could be explained by the greater presence of medical students in our sample, who are more willing to vaccinate. In this context, the risk perception may increase the willingness to get vaccinated, especially in medical students, as a higher self-perceived risk of contracting influenza is associated with better adherence to flu vaccination [10].

An even greater proportion of students would undertake COVID-19 vaccination once becomes available. In particular, as to flu vaccination, we found that being a Medicine and Surgery student and having undertaken a previous flu vaccination significantly predicts the intention of being vaccinated against influenza. Similarly, general concerns about the pandemic, increase of positive cases, increase in the number of deaths, as well as the distance to maintain with colleagues, the impossibility of attending university, and the impossibility of seeing friends were significantly associated with the intention to be vaccinated against flu. In parallel, those who declared to be more willing to undertake flu vaccination would also be significantly more willing to get vaccinated against COVID-19.

Recently a multicentric survey, conducted across 6 countries addressed to caregivers of children and adolescents aged from 1 to 19 years of age, reported an increase of 15.9% in the intentions to vaccinate children with respect to the previous year [21]. As to the COVID-19 vaccine, a multicentric survey conducted in July 2020 on 13,426 subjects from 19 countries, reported that 71.5% of respondents were very or somewhat likely to get vaccinated against COVID-19 [22] while considering a sample of 968 Italian citizens, 59% of the respondents reported to be likely to vaccinate for COVID-19 [16].

Overall, young adults, although less likely experience severe symptoms or death [23], are infectious if they acquire the SARS-CoV-2, and since they are internationally recognized as a group with potentially low compliance with public health measures, because of gatherings, and they maintain large and active social lives [24,25,26], they can play an important part in the spread of the virus in the community [27,28].

However, accumulating evidence shows that the level of concern and perceived vulnerability as occurring during the COVID-19 pandemic are a motivating factor to undertake a vaccine and, in particular, the flu vaccine [20]. In fact, the perceived level of a health threat is a strong predictor of people’s intention to adopt preventive behaviors, including undertaking flu vaccination [29,30,31]. On the other hand, however, emotional distress levels linked to health threats are voluble and changing over time and cannot guarantee that intentions to behave in a preventive manner will concretely translate into a concrete enactment of preventive behaviors. This implies that future educational flu vaccination campaigns should particularly rely on increasing individual literacy about the flu vaccine, reassuring on vaccine effectiveness and safety as well as sustaining a cultural change towards an engaged and aware approach to self-health management in order to guarantee adequate levels of population immunization against the risk of infectious diseases [32]. Furthermore, sensitizing campaigns to increase vaccination uptake should also focus on the social consequences of not getting vaccinated, since the role of pro-social messages in public health communication was widely demonstrated as a persuasive means for delivering preventive messages, particularly during the COVID-19 pandemic [33].

In interpreting the results of our survey, we acknowledge its main limitations: sampling was opportunistic so that we cannot infer the results to the entire Italian university student population; results might overestimate the willingness to undertake flu vaccination as medical students are a highly recommended target by the national immunization plan; lastly, we missed information from non-respondents.

Nonetheless, to our knowledge, this is the first study that investigates the impact of the COVID-19 pandemic on flu and COVID-19 vaccination intentions among university students and the possible influencing factors. Results show that university students represent a highly motivated population to undertake influenza vaccination and, even more so, the COVID-19 vaccination when made available. This trend needs to be better assessed over time as it could result from momentary concerns and vulnerability of the responders. For this reason, public health activities should keep their focus on cultural actions for students of all grades at multiple times, as engaging them at an earlier life stage can benefit themselves and, consequently, the whole society.

## 5. Conclusions

Flu vaccination represents, especially during the COVID-19 pandemic, a key tool to reduce mortality, morbidity and hospitalizations due to influenza. In particular, the involvement of the entire population could significantly contribute to reduce the spread of the influenza virus during the pandemic period. In this context, it is important to consider people and students’ willingness and intent to get vaccinated. Among these latter, being a medical student, having undertaken a previous vaccination against flu and having a high level of concern and perceived vulnerability to the COVID-19 pandemic positively predicts the intention to undertake flu vaccination. Moreover, understanding vaccination intentions for COVID-19 among students is also important as it helps to ensure adequate levels of population immunization against COVID-19. Indeed, students have an active social live based on relationships and, in this context, their vaccination uptake could play an important role in fighting the spread of the virus in the community. In this context, our study highlights that students are strongly involved and are willing to undertake both vaccinations.

## Figures and Tables

**Table 1 vaccines-09-00070-t001:** Predictors of flu and COVID-19 vaccination (adjusted OR: odds ratio; CI: 95% confidence interval).

**Willingness to get vaccinated**					
**Variable**	**Category**	**Flu Vaccination**			
		Yes	No	OR (95% CI)	*p*-value
		N (%)	N (%)		
Total		338 (77.52)	98 (22.48)		
Gender	Female	231 (75.24)	76 (24.76)	-	
	Male	107 (82.95)	22 (17.05)	1.31 (0.75–2.28)	0.34
		Median (IQR)	Median (IQR)		
Age		23.0 (22–25)	23.0 (21–25)	0.98 (0.94–1.03)	0.45
Faculty	Other	44 (59.46)	30 (40.54)	-	-
	Medicine	241 (87.96)	33 (12.04)	4.89 (2.70–8.85)	**<0.0001**
	Psychology	53 (60.23)	35 (39.77)	0.99 (0.53–1.85)	
Campus	(Brescia, Milano, Piacenza-Cremona)	92 (58.97)	64 (41.03)	-	
	Rome	246 (87.86)	34 (12.14)	4.95 (3.05–8.06)	**<0.0001**
Understand preventive measures	Strongly disagree	0	0	-	
	Disagree	1 (50.00)	1 (50.00)	1.53 (1.02–2.29)	**0.038**
	Moderate	7 (50.00)	7 (50.00)		
	Agree	102 (79.07)	27 (20.93)		
	Strongly agree	228 (78.35)	63 (21.65)		
Concern about the COVID-19 pandemic	No	90 (67.67)	43 (32.33)	-	
	Yes	243 (81.81)	54 (18.19)	1.13 (0.99–1.30)	**0.069**
Fear about the increase in deaths	Strongly disagree	4 (44.44)	5 (55.56)	-	
	Disagree	23 (65.71)	12 (34.29)	1.35 (1.05–1.74)	**0.019**
	Moderate	61 (78.21)	17 (21.79)		
	Agree	120 (80.00)	30 (20.00)		
	Strongly agree	130 (79.27)	34 (20.73)		
Fear about the increase in positive cases	Strongly disagree	5 (41.67)	7 (58.33)	-	
	Disagree	28 (68.29)	13 (31.71)	1.44 (1.14–1.83)	**0.003**
	Moderate	67 (75.28)	22 (24.72)		
	Agree	130 (81.76)	29 (18.24)		
	Strongly agree	108 (80.00)	27 (20.00)		
Suffering from distance to fellow students	Strongly disagree	37 (75.51)	12 (24.49)	-	
	Disagree	39 (70.91)	16 (29.09)	1.21 (1.01–1.46)	**0.035**
	Moderate	53 (68.83)	24 (31.17)		
	Agree	117 (79.05)	31 (20.95)		
	Strongly agree	92 (85.95)	15 (14.02)		
Suffering from the impossibility of attending university	Strongly disagree	35 (79.55)	9 (20.45)	-	
	Disagree	31 (64.58)	17 (35.42)	1.20 (1.01–1.44)	**0.042**
	Moderate	40 (71.43)	16 (28.57)		
	Agree	117 (78.52)	32 (21.48)		
	Strongly agree	115 (82.73)	24 (17.27)		
Suffering from distance to friends	Strongly disagree	11 (68.75)	5 (31.25)	-	
	Disagree	16 (50.00)	16 (50.00)	1.31 (1.07–1.60)	**0.009**
	Moderate	49 (75.38)	16 (24.62)		
	Agree	101 (81.45)	23 (18.55)		
	Strongly agree	161 (80.90)	38 (19.10)		
Desire to contribute much more to facing the pandemic	Strongly disagree	10 (61.50)	6 (37.50)	-	
	Disagree	31 (70.45)	13 (29.55)	1.22 (0.98–1.53)	0.081
	Moderate	84 (70.59)	35 (29.41)		
	Agree	124 (82.12)	27 (17.88)		
	Strongly agree	89 (83.96)	17 (16.04)		
Previous flu vaccination	No	128 (61.54)	80 (38.46)	-	
	Yes	210 (92.11)	18 (7.89)	5.82 (3.27–10.37)	**<0.0001**
COVID-19 vaccine intention	No	16 (32.00)	34 (68.00)	-	
	Yes	322 (83.42)	64 (16.58)	9.58 (4.78–19.19)	**<0.0001**
**Willingness to get vaccinated**					
**Variable**	**Category**	**COVID-19 vaccination**			
		Yes	No	OR (95% CI)	*p*-value
		N (%)	N (%)		
Total		386 (88.53)	50 (11.47)		
Gender	Female	266 (86.64)	41 (13.36)	-	
	Male	120 (93.02)	9 (6.98)	1.82 (0.85–3.92)	0.12
		Median (IQR)	Median (IQR)		
Age		23.0 (22–25)	23.0 (21–25)	0.99 (0.94–1.05)	0.657
Faculty	Other	66 (89.19)	8 (10.81)	-	-
	Medicine	254 (92.70)	20 (7.30)	1.57 (0.66–3.76)	0.306
	Psychology	66 (75.00)	22 (25.00)	0.38 (0.15–1.01)	0.051
Campus	(Brescia, Milano, Piacenza-Cremona)	128 (82.05)	28 (17.95)	-	
	Rome	258 (92.14)	22 (7.86)	2.48 (1.34–4.57)	**0.004**
Understand preventive measures	Strongly disagree	0 (00.00)	0 (00.00)	-	
	Disagree	1 (50.00)	1 (50.00)	1.66 (0.97–2.83)	0.064
	Moderate	8 (57.14)	6 (42.86)		
	Agree	119 (92.25)	10 (7.75)		
	Strongly agree	258 (88.66)	33 (11.34)		
Concern about the COVID-19 pandemic	No	29 (58.00)	21 (42.00)	-	
	Yes	268 (70.53)	112 (29.47)	1.18 (0.99–1.40)	0.068
Fear about the increase in deaths	Strongly disagree	4 (44.44)	5 (55.56)	-	
	Disagree	29 (82.86)	6 (17.14)	1.68 (1.24–2.66)	**0.001**
	Moderate	66 (84.62)	12 (15.38)		
	Agree	138 (92.00)	12 (8.00)		
	Strongly agree	149 (90.85)	15 (9.15)		
Fear about the increase in positive cases	Strongly disagree	6 (05.00)	6 (50.00)	-	
	Disagree	36 (87.80)	5 (12.20)	1.58 (1.18–2.10)	**0.002**
	Moderate	75 (84.27)	14 (15.73)		
	Agree	147 (92.45)	12 (7.53)		
	Strongly agree	122 (90.37)	13 (9.63)		
Suffering from distance to fellow students	Strongly disagree	41 (83.67)	8 (16.33)	-	
	Disagree	44 (80.00)	11 (20.00)	1.23 (0.97–1.55)	0.084
	Moderate	66 (89.61)	8 (10.39)		
	Agree	136 (91.89)	12 (8.11)		
	Strongly agree	96 (89.72)	11 (10.28)		
Suffering from the impossibility of attending university	Strongly disagree	39 (88.64)	5 (11.36)	-	
	Disagree	39 (81.25)	9 (18.75)	1.06 (0.84–1.35)	0.6
	Moderate	52 (92.86)	4 (7.14)		
	Agree	133 (89.26)	16 (10.74)		
	Strongly agree	123 (88.49)	16 (11.51)		
Suffering from distance to friends	Strongly disagree	10 (62.50)	6 (37.50)	-	
	Disagree	29 (90.63)	3 (9.38)	1.30 (1.01–1.67)	**0.045**
	Moderate	56 (86.15)	9 (13.85)		
	Agree	110 (88.71)	14 (11.29)		
	Strongly agree	181 (90.95)	18 (9.05)		
Desire to contribute much more to facing the pandemic	Strongly disagree	11 (68.75)	5 (31.25)	-	
	Disagree	39 (88.64)	5 (11.36)	1.24 (0.91–1.68)	0.178
	Moderate	104 (87.39)	15 (12.61)		
	Agree	136 (90.07)	15 (9.93)		
	Strongly agree	96 (90.57)	10 (9.43)		
Previous flu vaccination	No	174 (83.65)	34 (16.35)	-	
	Yes	212 (92.98)	16 (7.02)	2.06 (1.07–3.97)	**0.032**

In bold statistically significant results.

## Data Availability

Data presented in this study are available upon request from the corresponding author. Data are not publicly available as they are property of the Università Cattolica del Sacro Cuore, Rome Italy.

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
