# Peer review of "Impact of COVID-19 Pandemic on Flu and COVID-19 Vaccination Intentions among University Students"

_vaccines, 2021, doi:10.3390/vaccines9020070_

Round 1

Reviewer 1 Report

Dear authors, the topic treated is very actual, going to face two important challenges that public health must face today, vaccination hesitancy (also present among healthcare workers) and the COVID-19.
Here are some comments to make improvements to the final draft that authors may decide to make:

  • Introduction:
    • explain the meaning of vaccination hesitancy in the introduction section and also the meaning of risk perception that could be correlate with the impact on vaccine uptake (discuss it into discussion section)
  • Results:
    • Indicate in the results section if the authors talk about unadjusted or adjusted ORs and insert the observed p-value were request.
    • The authors stated “We collected a total of 436 questionnaires” Please indicate response rate. How many students attendind investigated universities?
    • The authors indicated “undergraduate students attending one of the four campuses of Università Cattolica del Sacro Cuore (Milan, Lombardy; Brescia, Lombardy; Piacenza-Cremona, Lombardy and Emilia-Romagna; Rome, Lazio) located in 3 Italian regions. The faculties involved were: Medicine and Surgery(Rome), Psychology (Milan and Brescia), Economy (Rome), Economy and Law (Piacenza-Cremona), Agricultural, Food, and Environmental Sciences (Piacenza-Cremona), Bank-ing, Financial, and Insurance Sciences (Milan), and Education Sciences (Milan).”Please clarify this point.
    • In results section insert table and figure to improve readability of the article, also add number of subjects near to percentage or percentage near number.
    • It could be interesting to compare the compliance of medical and not medical students. Are there any differences?
  • Discussion:
    • Improve discussion section: i.e. authors could indicate the role of young into spread of COVID-19, the role of risk perception (in results section the authors report: Concerning perception and concerns on COVID-19 pandemic, over 95% of the respondents reported to understand the relevance of lockdown’s measures adopted in the first wave of pandemic in Italy, in terms of pandemic containment. More than 67% referred to fear about the increase in positive cases and 70% were concerned about the reported number of deaths. In general, almost 40% of students reported great concern about the containment of the pandemic in Italy, and around 60% of the students were willing to contribute more to face the pandemic. Concerning the students’ social life, 59%, 58% and 65% suffered from the impossibility of seeing friends, of seeing colleagues and of attending the university, respectively) into vaccine uptake. 
    • Add references in discussion section. There are many studies in Italy that evaluated the willingness to get vaccine in medical and not medical students first to this one.

Finally, I am very happy to have read this original article. 
kind regards

Author Response

REVIEWER 1

Dear authors, the topic treated is very actual, going to face two important challenges that public health must face today, vaccination hesitancy (also present among healthcare workers) and the COVID-19.
Here are some comments to make improvements to the final draft that authors may decide to make:

We really thank the reviewer for all their valuable comments that have helped us to improve our manuscript that now could be suitable for publication.

  • Introduction: explain the meaning of vaccination hesitancy in the introduction section and also the meaning of risk perception that could be correlate with the impact on vaccine uptake (discuss it into discussion section)

Thank you for your comment. We added this part in the manuscript. We hope we have properly addressed the issue raised by the reviewer. In particular: “Nevertheless, over the last decades, opposition to vaccination got amplified and the vaccine hesitancy phenomenon arose. This phenomenon is defined as a set of beliefs, attitudes and behaviors (alone or in combination), exhibited by common people and, in some cases, also by healthcare professionals, in regard to their own or their children’s immunizations, causing a decrease of vaccine coverage and an increasing risk of preventable disease epidemics (8). In this context, the fear of potential damages after receiving vaccine raises concern among people possibly leading to a reduction in the vac-cine uptake (9). In addition, self-perceived risk (both influenza and COVID-19) may impact in the willingness to vaccine uptake (10).”

  • Results:
    • Indicate in the results section if the authors talk about unadjusted or adjusted ORs and insert the observed p-value were request.

Thank you. All ORs are adjusted (we corrected this part in the main text). We added a table with all information requested.

    • The authors stated “We collected a total of 436 questionnaires” Please indicate response rate. How many students attending investigated universities?

Thank you. We administered the questionnaire online on the University's intranet page. For privacy reasons, in fact, our Ethical Committee does not allow surveys to be sent to personal emails. For this reason, we can calculate the response rate with respect to the number of people who accessed the page, assuming that these people are those aware of the questionnaire. In particular, the access mode was structured in such a way that from each personal page the survey could be completed only once. In this way, each access corresponds to only one student (whether they completed the questionnaire or not). The response rate is 436/559, equal to 78%.

    • The authors indicated “undergraduate students attending one of the four campuses of Università Cattolica del Sacro Cuore (Milan, Lombardy; Brescia, Lombardy; Piacenza-Cremona, Lombardy and Emilia-Romagna; Rome, Lazio) located in 3 Italian regions. The faculties involved were: Medicine and Surgery (Rome), Psychology (Milan and Brescia), Economy (Rome), Economy and Law (Piacenza-Cremona), Agricultural, Food, and Environmental Sciences (Piacenza-Cremona), Banking, Financial, and Insurance Sciences (Milan), and Education Sciences (Milan).” Please clarify this point.

Thank you for you feedback on this point. We modified the manuscript and we hope that it is clearer now. In particular, Università Cattolica del Sacro Cuore has 4 campuses in Italy in the cities of Milan, Brescia, Piacenza-Cremona and Rome. These 4 campuses are in 3 different Italian regions (2 campuses are located in Lombardy, 1 campus is located partly in Lombardy and partly in Emilia-Romagna and 1 campus is located in Lazio). In particular, although they are very close cities (<35 km), Piacenza and Cremona belong to two different regions. The campus is located partly in Piacenza and partly in Cremona. For this reason, the campus is called Piacenza-Cremona and, although unique, is located in two different regions. In conclusion:

1 campus in Milan (Lombardy)

1 campus in Brescia (Lombardy)

1 campus in Rome (Lazio)

1 campus partly in Piacenza (Emilia-Romagna) and partly in Cremona (Lombardy).

    • In results section insert table and figure to improve readability of the article, also add number of subjects near to percentage or percentage near number.

Thank you, we added the table and we added the number of subjects near to percentage in the main text.

    • It could be interesting to compare the compliance of medical and not medical students. Are there any differences?

We adjusted the OR for gender and campuses. Moreover, we found that the willingness to undertake flu vaccination is associated with the intention to get vaccinated against COVID-19 without differences among the faculties. We added the table where we have shown all our results.

  • Discussion:
    • Improve discussion section: i.e. authors could indicate the role of young into spread of COVID-19, the role of risk perception (in results section the authors report: Concerning perception and concerns on COVID-19 pandemic, over 95% of the respondents reported to understand the relevance of lockdown’s measures adopted in the first wave of pandemic in Italy, in terms of pandemic containment. More than 67% referred to fear about the increase in positive cases and 70% were concerned about the reported number of deaths. In general, almost 40% of students reported great concern about the containment of the pandemic in Italy, and around 60% of the students were willing to contribute more to face the pandemic. Concerning the students’ social life, 59%, 58% and 65% suffered from the impossibility of seeing friends, of seeing colleagues and of attending the university, respectively) into vaccine uptake. 

Thank you for the suggestion, we have made changes. In particular:

“Therefore, despite the fact that, in general, students are a population that presents a low flu vaccination intention, the pandemic may have increased the willingness to vaccinate. It has already been observed, in fact, an intention to vaccinate that reaches al-most 60% in the student population for the 2020-21 campaign (20), a figure higher than the average values generally observed at the national level. Our study shows a slightly higher willingness to vaccinate (77%) that could be explained by the greater presence of medical students in our sample, who are more willing to vaccinate. In this context, the risk perception may increase the willingness to get vaccinated, especially in the medical students, as a higher self-perceived risk of contracting influenza is associated with a better adherence to flu vaccination (10).”

    • Add references in discussion section. There are many studies in Italy that evaluated the willingness to get vaccine in medical and not medical students first to this one.

Thank you. We added references in the discussion.

Reviewer 2 Report

The paper entitled “IMPACT OF COVID-19 PANDEMIC ON FLU AND COVID-19 VACCINATIONS’ INTENTIONS AMONG UNIVERSITY STUDENTS” reports some interesting results and may be of interest for the Vaccines readers. I have only few comments that are listed below.

  • In the Methods it is stated that both uni- and multivariable logistic models were used. However, the ORs provided through the Results section do not distinguish between unadjusted and adjusted ORs. Please, differentiate them (e.g. OR vs aOR).
  • In the Results please provide the response rate.
  • In the Results there are three phrases stating a “borderline association”. It is not recommended to use words such as “borderline/barely significant”, “trend to significance”, etc. You may state, for example, “Males had a higher intention to undertake flu vaccination (OR 1.60, 95% CI [0.94-2.71]) although the alpha did not reach <0.05 [Insert the observed p-value].
  • The previous influenza vaccine coverage rate seems high. I can understand a rate of 76% among the medical students (although is much higher than the reported rate among the Italian HCPs in general). On the other hand, from the numbers provided the coverage tare appears to be 34% (55/162) among the students from other faculties. According to the ISTAT Health for All database, the latest available estimate for people aged 18-49 years is around 2-3%. How would you interpret this discrepancy? Social desirability bias on the wave of COVID-19 pandemic? Positive response bias (people more interested in influenza vaccination were more likely to participate)? Selection bias (only university students were recruited)?

Author Response

REVIEWER 2

The paper entitled “Impact of covid-19 pandemic on flu and covid-19 vaccinations’ intentions among university students” reports some interesting results and may be of interest for the Vaccines readers. I have only few comments that are listed below.

We really thank the reviewer for all their valuable comments that have helped us to improve our manuscript that now could be suitable for publication.

  • In the Methods it is stated that both uni- and multivariable logistic models were used. However, the ORs provided through the Results section do not distinguish between unadjusted and adjusted ORs. Please, differentiate them (e.g. OR vs aOR).

Thank you. All ORs are adjusted (we specify this concept). We added a table with all information requested

  • In the Results please provide the response rate.

Thank you. We administered the questionnaire online on the University's intranet page. For privacy reasons, in fact, our Ethical Committee does not allow surveys to be sent to personal emails. For this reason, we can calculate the response rate with respect to the number of people who accessed the page, assuming that these people are those aware of the questionnaire. In particular, the access mode was structured in such a way that from each personal page the survey could be completed only once. In this way, each access corresponds to only one student (whether they completed the questionnaire or not). The response rate is 436/559, or 78%.

  • In the Results there are three phrases stating a “borderline association”. It is not recommended to use words such as “borderline/barely significant”, “trend to significance”, etc. You may state, for example, “Males had a higher intention to undertake flu vaccination (OR 1.60, 95% CI [0.94-2.71]) although the alpha did not reach <0.05 [Insert the observed p-value].

Thank you for the advice, we changed the text. We also added a table with all information requested.

  • The previous influenza vaccine coverage rate seems high. I can understand a rate of 76% among the medical students (although is much higher than the reported rate among the Italian HCPs in general). On the other hand, from the numbers provided the coverage tare appears to be 34% (55/162) among the students from other faculties. According to the ISTAT Health for all database, the latest available estimate for people aged 18-49 years is around 2-3%. How would you interpret this discrepancy? Social desirability bias on the wave of COVID-19 pandemic? Positive response bias (people more interested in influenza vaccination were more likely to participate)? Selection bias (only university students were recruited)?

Thank you. We added this part in the introduction “Nevertheless, over the last decades, opposition to vaccination got amplified and the vaccine hesitancy phenomenon arose. This phenomenon is defined as a set of beliefs, attitudes and behaviors (alone or in combination), exhibited by common people and, in some cases, also by healthcare professionals, in regard to their own or their children’s immunizations, causing a decrease of vaccine coverage and an increasing risk of preventable disease epidemics (8). In this context, the fear of potential damages after receiving vaccine raises concern among people possibly leading to a reduction in the vac-cine uptake (9). In addition, self-perceived risk (both influenza and COVID-19) may impact in the willingness to vaccine uptake (10).”

And this part in the discussion:

“Therefore, despite the fact that, in general, students are a population that presents a low flu vaccination intention, the pandemic may have increased the willingness to vaccinate. It has already been observed, in fact, an intention to vaccinate that reaches al-most 60% in the student population for the 2020-21 campaign (20), a figure higher than the average values generally observed at the national level. Our study shows a slightly higher willingness to vaccinate (77%) that could be explained by the greater presence of medical students in our sample, who are more willing to vaccinate. In this context, the risk perception may increase the willingness to get vaccinated, especially in the medical students, as a higher self-perceived risk of contracting influenza is associated with a better adherence to flu vaccination (10).”

Thank you for all the comments

Reviewer 3 Report

This paper is important to understanding vaccine hesitancy in a key demographic of COVID transmission. 

I have the following observations about the analysis- it seems to me that the population and demographics is known for the complete population of students- so why haven't the estimates been weighted against the known population to unbias the final estimates?

Similar to the point above the final regression analyses should be properly adjusted for the population sampled to produce estimates less biased by the convenience sample.

The ability to weight relative to the known population is a key strength of this study design and once the estimates are adjusted properly I think this will be a good paper. 

Author Response

This paper is important to understanding vaccine hesitancy in a key demographic of COVID transmission. 

We really thank the reviewer for all their valuable comments that have helped us to improve our manuscript that now could be suitable for publication.

I have the following observations about the analysis

  • it seems to me that the population and demographics is known for the complete population of students- so why haven't the estimates been weighted against the known population to unbias the final estimates? Similar to the point above the final regression analyses should be properly adjusted for the population sampled to produce estimates less biased by the convenience sample. The ability to weight relative to the known population is a key strength of this study design and once the estimates are adjusted properly I think this will be a good paper.

We modified the analysis applying univariable and multivariable survey weighted logistic regressions to assess the influence of independent variables on each binary outcome investigated (with the results expressed as odds ratios [OR], 95% CI). In order to account for underrepresented groups in the student’s population, poststratification has been applied for adjusting the sampling weights. The poststratification has been done using the known distribution of gender of the UCSC students (http://ustat.miur.it/dati/didattica/italia/atenei-non-statali/milano-cattolica).